# Isopeptide Bonding *In Planta* Allows Functionalization of Elongated Flexuous Proteinaceous Viral Nanoparticles, including Non-Viable Constructs by Other Means

**DOI:** 10.3390/v15020375

**Published:** 2023-01-28

**Authors:** Daniel A. Truchado, Sara Rincón, Lucía Zurita, Flora Sánchez, Fernando Ponz

**Affiliations:** Centro de Biotecnología y Genómica de Plantas (CBGP) (UPM-INIA/CSIC), Autopista M40, km 38, Campus Montegancedo, Pozuelo de Alarcón, 28223 Madrid, Spain

**Keywords:** potyvirus, functionalization, VLPs, Spytag/SpyCatcher, VIP

## Abstract

Plant viral nanoparticles (VNPs) have become an attractive platform for the development of novel nanotools in the last years because of their safety, inexpensive production, and straightforward functionalization. Turnip mosaic virus (TuMV) is one example of a plant-based VNP used as a nanobiotechnological platform either as virions or as virus-like particles (VLPs). Their functionalization mainly consists of coating their surface with the molecules of interest via chemical conjugation or genetic fusion. However, because of their limitations, these two methods sometimes result in non-viable constructs. In this paper, we applied the SpyTag/SpyCatcher technology as an alternative for the functionalization of TuMV VLPs with peptides and proteins. We chose as molecules of interest the green fluorescent protein (GFP) because of its good traceability, as well as the vasoactive intestinal peptide (VIP), given the previous unsuccessful attempts to functionalize TuMV VNPs by other methods. The successful conjugation of VLPs to GFP and VIP using SpyTag/SpyCatcher was confirmed through Western blot and electron microscopy. Moreover, the isopeptide bond between SpyTag and SpyCatcher occurred in vivo in co-agroinfiltrated *Nicotiana benthamiana* plants. These results demonstrated that SpyTag/SpyCatcher improves TuMV functionalization compared with previous approaches, thus implying the expansion of the application of the technology to elongated flexuous VNPs.

## 1. Introduction

Viral-derived nanoparticles (VNPs), including virions and empty virus-like particles (eVLPs), have increasingly made their way to become a center player in the development of nanobiotechnology, through their functionalization with multiple purposes [1]. Most VNP functionalizations are approached either through genetic conjugation of the viral coat protein (CP) with nucleic acid sequences encoding a foreign peptide/protein in order to generate a CP-based fusion protein, or by in vitro chemical conjugations (including encapsidations) of the purified particles with the desired functionalizing molecule. Using this last approach, non-proteinaceous molecules can be incorporated.

In the case of plant VNPs, the *in planta* production of genetically functionalized particles can be carried out using molecular farming, thus taking advantage of the several benefits of plants as biofactories. Having the functionalized VNPs made within inoculated plants facilitates purification of the final nanoparticle to be readily used for the desired application [2]. Until recently, this was only possible through inoculation of the host plant with an infectious clone of the recombinant virus carrying the genetically produced fusion CP protein, for the case of virions. In the case of VLPs, agroinoculation of an expression vector carrying the construct of the fusion CP allows its production in the recipient plant cells, and its assembly into particles, if these are viable.

Most examples of the plant-made functionalized VNPs can be found within icosahedral viruses with spherical symmetry. Among these, cowpea mosaic virus has been the most exploited, although some others have also been used [3]. In comparison with spherical viruses, elongated ones with a helical symmetry can represent a better choice in some instances, as their number of subunits per particle is much larger. Among these, tobacco mosaic virus (TMV) and potato virus X (PVX) have been widely used [4]. In our own work, we have a long track record in the development of VNPs from turnip mosaic virus (TuMV) for multiple nanobiotechnological applications [5,6,7,8,9,10,11,12,13,14,15]. TuMV is a potyvirus, flexuous elongated viruses, whose long virions (750 nm long and 13.5 nm wide) are made of over 2000 CP molecules, and of which eVLPs can be formed *in planta* [16]. TuMV VNPs are a good alternative to those derived from other more popular elongated viruses (such as TMV and PVX) because they not only have the advantages of flexuous viruses, but are also longer. This feature makes it possible to potentially bind more molecules of interest per VNP. So far, TuMV eVLPs were functionalized via genetic fusion, where the protein of interest was expressed *in planta* together with the CP, with or without a flexible linker in between [11]. Genetic fusion allows the functionalization of all CP available in the eVLPs, which, in the case of TuMV, is approximately 2000 copies of the protein of interest per eVLP. Despite these promising data, this approach may not be applicable in all cases, as steric hindrances caused by large proteins of interest or other structural traits of the functionalizing molecule may impede the self-assembly of the CPs. One solution to get round this problem is the partial functionalization of the VNPs, where their self-assembly is possible because there is a certain percentage of non-functionalized CPs [17]. Other options to avoid this problem are the addition of linkers between the capsid protein and the protein of interest or the deletion of some amino acids of the capsid protein to allow the fusion of larger peptides [4].

As an alternative to genetic fusion, a relatively recent method has been successfully applied for a complete functionalization of VNPs: SpyTag/SpyCatcher technology, which implies a gene fusion and a subsequent chemical conjugation [18]. SpyTag/SpyCatcher is based on a spontaneous intramolecular isopeptide bond occurring in the CnaB2 domain of fibronectin-binding protein in *Streptococcus pyogenes*, more specifically, between the amine of Lys^31^ and the carboxyl group of Asp^117^. This reaction takes place under a wide range of environmental conditions, and it is irreversible. By splitting the CnaB2 domain, two peptides were obtained: SpyTag (the 13 amino acids of the C-terminal region carrying the reactive Asp) and SpyCatcher (the rest of the domain which will eventually bind to its partner). Through the gene fusion of SpyTag and SpyCatcher to either the N-terminus or C-terminus of two proteins of interest, these two proteins will be finally joined covalently to each other to a high yield [18,19]. Recently, SpyTag/SpyCatcher technology has been applied for virus nanoparticle functionalization in different plant viruses [20,21], with foreign antigen display being the most common use in general [22,23,24,25]. However, to the best of our knowledge, potyvirus VNPs have never been functionalized through this method.

In this study, we compare the genetic fusion and the SpyTag/SpyCatcher technology to functionalize TuMV VLPs with two different proteins: the green fluorescent protein (GFP) and the vasoactive intestinal peptide (VIP). GFP was chosen as a model protein thanks to its good traceability and because it had been previously selected to successfully decorate VLPs using SpyTag/SpyCatcher technology [20]. On the other hand, VIP was selected thanks to its important role as an anti-inflammatory and immunomodulator in humans, which makes it a key element in inflammatory and autoimmune diseases [26]. Free VIP is currently used as a treatment for some of these conditions because of its protective effect, although its quick degradation inside the body means these treatments are far from its full potential [27]. By conjugating VIP to TuMV VLPs, the integrity of the peptide should be greater and, perhaps, will improve its administration. Attempts to functionalize TuMV VNPs with VIP were carried out previously via chemical conjugation by our group, although it took place with low efficiency. The functionalization with VIP using genetic fusion turned out unsuccessful [28]. Thus, our main objective was to compare two different functionalization techniques in order to improve the coating of TuMV VLPs with a protein of interest. To that end, we assessed the suitability of gene fusion as a tool for the functionalization of TuMV with GFP and VIP in the first place. In parallel, we tested whether SpyTag/SpyCatcher technology could be applied to TuMV VNPs using the GFP as a model antigen. Furthermore, we tested if the spontaneous isopeptide bond between SpyTag and SpyCatcher in the VLP-GFP construct would take place in vivo in *Nicotiana benthamiana* plants after agroinfiltration. Finally, we functionalized TuMV VLPs with VIP using SpyTag/SpyCatcher to evaluate if this technology had a better performance than gene fusion and chemical conjugation with this peptide and, therefore, could be considered as a good alternative to functionalize TuMV VNPs with proteinaceous molecules in the future.

## 2. Materials and Methods

### 2.1. Cloning in Expression Vectors and Agroinfiltration

Eight different synthetic genes were designed and ordered from GeneArt (Thermo Fisher Scientific, Regensburg, Germany). Six of the constructs resulted from all the possible combinations between SpyTag and SpyCatcher with TuMV CP, GFP, and VIP, whereas the other two consisted of GFP and VIP nucleotide sequences fused to the sequence encoding the CP. All proteins of interest were fused in the N-terminus of TuMV CP as this is the part projected to the exterior of the VLP (Figure 1). These gene constructs were cloned in the pEAQ-HT-DEST1 expression vector and *E. coli* top 10 cells (ThermoFisher Scientific) were used to propagate the plasmids. After confirmation by Sanger sequencing, *Agrobacterium tumefaciens* (LBA4404 strain) cultures were transformed with each construct individually. For transient expression of the constructs, we agroinfiltrated *Nicotiana benthamiana* plants using a needleless syringe following a previously described protocol [29]. Briefly, *A. tumefaciens* strains were subcultured overnight, pelleted, and resuspended in MMA (10 mM MES buffer, pH 5.6; 10 mM magnesium chloride; 100 µM acetosyringone) until a OD_600_ = 1.2 was reached. Then, each construct was agroinfiltrated alone and, in those with SpyTag/SpyCatcher, co-agroinfiltrated with its complementary construct. For co-agroinfiltrations, the two *A. tumefaciens* cultures carrying the complementary constructs were mixed with the same concentration of each strain before infiltration in the leaves of *N. benthamiana*.

### 2.2. Protein Extraction and Monitoring of GFP Formation

We monitored the production of GFP in the corresponding plants by illuminating the leaves using UV light in a dark room. For all constructs, agroinfiltrated tissue was harvested 7 dpi. Standard 44 mm circles of agroinfiltrated leaves (≈75 mg) were cut and blended immediately in protein extraction buffer (50 mM Tris-HCl, pH 7.25; 150 mM NaCl; 2 mM EDTA; 0.1% (*v*/*v*) Triton X-100; and 1× SIGMAFAST^TM^ protease inhibitor) using a tissue grinder. Crude extracts were stored at −80 °C until use.

### 2.3. VLPs Characterization

Characterization of assembled VLPs was performed by SDS-PAGE, Western blot, ELISAs, and transmission electron microscopy (TEM, ICTS-CNME, Madrid, Spain).

Proteins from the crude extracts were separated in 12% SDS-PAGE (with a stacking gel of 4%) and transferred to a PVDF membrane (Amersham^TM^ Hybond^TM^ P 0.45 PVDF, Cytiva, Cornellá de Llobregat, Spain). The membrane was blocked afterwards with 2% skimmed milk in PBS. CP detection was carried out through incubation with the primary monoclonal antibody anti-POTY (Agdia, Elkhart, IN, USA) diluted 1:200 and the subsequent treatment with the secondary antibody anti-mouse (Agdia) conjugated to alkaline phosphatase diluted 1:500. The presence of GFP was detected by incubating the membrane with the primary antibody anti-GFP (Clontech, Mountain View, CA, USA) diluted 1:1000, followed by the binding of the secondary antibody anti-mouse (Agdia) conjugated to alkaline phosphatase diluted 1:500. The detection of VIP was performed by incubating the membrane with the primary antibody anti-VIP (Invitrogen, Waltham, MA, USA) diluted 1:1000, followed by a treatment with the secondary antibody anti-rabbit (Invitrogen) conjugated to alkaline phosphatase diluted 1:2000.

To detect the presence of CP and VIP in the corresponding VLPs, indirect enzyme-linked immunosorbent assays (ELISAs) were carried out. High binding plates (Fisher Scientific, Waltham, MA, USA) were coated with 5 µL of purified VLPs diluted with 195 µL of 50 mM sodium carbonate buffer, pH 9.6, and incubated overnight at 4 °C. Plates were incubated for 1 h at 37 °C with the corresponding primary antibody in PBS, 0.05% Tween 20, 2% PVP-40, 2 mg/mL BSA (anti-POTY diluted 1:200; anti-VIP diluted 1:2000). Then, an incubation with the corresponding alkaline phosphatase-conjugated secondary antibody was performed for 1 h at 37 °C. Alkaline phosphatase activity was detected using nitrophenylphosphate. The optical density of samples was measured at 405 nm (SPECTROstar Nano^®^; BMG Labtech, Ortenberg, Germany).

To check VLPs’ assembly, TEM was applied. Electron microscopy grids (400 mesh copper-carbon-coated) were floated at room temperature with a 10 μL drop of VLPs (1:25 dilution of the crude extracts in 50 mM borate buffer, pH 8.1). After 10 min, the grids were washed with five drops of distilled H_2_O for 5 min in each drop and incubated with 10 µL of anti-GFP or anti-VIP (both antibodies diluted 1:50 in 50 mM borate buffer, pH 8.1) for 10 min for decoration. No blocking step was carried out. Only in the case of VLPs-VIP, the grids were washed again in distilled H_2_O and then incubated for 10 min with 10 µL of a 5 nm gold-labelled anti-rabbit antibody for immunogold electron microscopy.

Finally, all the grids were rinsed five times with five drops of distilled H_2_O and stained with 2% uranyl acetate for 2 min. Samples were eventually examined on a transmission electron microscope (JEM JEOL 1400, Tokyo, Japan).

## 3. Results

### 3.1. Genetic Fusion

In the construct with GFP, Western blot analyses revealed the presence of the expected band of 61 kDa corresponding to the fusion of the CP and the GFP with the flexible linker (Figure 2A lane 2; and Figure 2B lane 7). Bands of a lower molecular weight were also observed, probably due to some kind of degradation of the recombinant protein in the process. In the case of CP-VIP, we obtained no result that proved the formation of a viable eVLP, as expected.

### 3.2. SpyTag/SpyCatcher

In the constructs SpyTag-GFP + SpyCatcher-CP and SpyCatcher-GFP + SpyTag-CP, the production of GFP and its correct folding was demonstrated visually by the differential fluorescence under UV light of the constructs with GFP compared with non-agroinfiltrated leaves (Figure 3).

Western blot analyses of protein extracts showed that all constructs with GFP were produced in plants when using SpyTag/SpyCatcher (Figure 2B). In the co-agroinfiltrated constructs, we observed bands of the expected size if the reaction between SpyTag and SpyCatcher took place (81.6 kDa for SpyTag-CP + SpyCatcher-GFP, and 75.2 kDa for SpyCatcher-CP + SpyTag-GFP). Two bands were observed in the constructs with SpyTag-GFP alone and SpyTag-GFP + SpyCatcher-CP, possibly due to an unknown reaction that cleaves SpyTag from GFP, as one band has the expected size of SpyTag-GFP (28.8 kDa) and the other has the size of GFP (27 kDa). The same occurred in the case of SpyCatcher-GFP, where we observed the same band of GFP alone and a band with the expected size for SpyCatcher-GFP (46.4 kDa). Moreover, we tried to mix in vitro the crude extracts of these constructs that were agroinfiltrated separately by placing Eppendorf tubes with the extracts in a mini shaker at room temperature for 5 min. However, Western blot analyses showed that in vitro binding of SpyTag and SpyCatcher was not successful under those conditions, as we could not observe any band with the expected size of 75.2 kDa (Figure 2B lane 3). We observed assembled eVLPs for both SpyTag-CP + SpyCatcher-GFP and SpyCatcher-CP + SpyTag-GFP in the micrographs obtained by TEM (Figure 4). In both constructs with GFP, the great length of the constructs is remarkable, a characteristic already observed in other TuMV eVLPs [13,16].

Regarding the constructs with VIP, Western blot analyses revealed that one of the constructs was produced in plants: SpyTag-CP + SpyCatcher-VIP (Figure 5). In this case, a band with the expected size (51.8 kDa) was found together with other bands of a higher molecular weight, likely corresponding to multimeric forms (Figure 5, lane 2). No band was found in the co-agroinfiltration where VIP was joined to SpyTag (Figure 5, lane 1).

The results obtained in the ELISAs showed that both the SpyTag-CP and the SpyCatcher-CP were positive when using anti-potyvirus as the primary antibody (Figure 6A). However, when using anti-VIP as the primary antibody, the co-agroinfiltration SpyTag-CP + SpyCatcher-VIP had higher absorbance values. SpyTag-VIP and SpyCatcher-VIP alone, together with the co-agroinfiltration SpyCatcher-CP + SpyTag-VIP, presented low absorbance values, although SpyCatcher-VIP had slightly higher results (Figure 6B).

Immunogold electron microscopy showed that only ST-CP + SC-VIP co-agroinfiltration was marked with colloidal gold (Figure 7). Unlike constructs with GFP, TuMV VLPs-VIP were shorter than free virions, something already observed in a construct functionalized with the allergen Pru p 3 [11].

## 4. Discussion

The functionalization of VNPs is a key step to create novel nanotools out of viral structural proteins. In the case of TuMV, the SpyTag/SpyCatcher technology represented the best alternative for potential extensive surface coverage with a protein of interest compared with genetic fusion, as it allows the coating of the VLPs while presenting no steric hindrances to the self-assembly of the capsid proteins.

Direct fusion of peptides or proteins via genetic fusion has long been used because it is a simple, effective, and selective method for functionalizing VLPs [4]. As in the case of GFP and VIP, the protein of interest is fused to a CP domain projected towards the exterior of the VLP. In TuMV, that region is the N-terminus in virions as well as in VLPs [16]. In the case of functionalization with GFP by genetic fusion, no steric hindrances were observed even though GFP could be considered as a relatively large protein (≈27 kDa) to be fused to the TuMV CP (≈33 kDa). The recombinant protein CP-GFP was detected through Western blot analyses, either with anti-potyvirus or with anti-GFP, showing that this construct was successfully produced in the plant. Surface display of GFP is not the only example of a successful functionalization of TuMV eVLPs through genetic fusion. Recently, our group developed another eVLP functionalized with the complete allergen Pru p 3 to test a potential novel food allergy treatment in allergic mice [11]. In this case, Pru p 3 was physically separated from the CP through a flexible linker to avoid undesired interactions as well. However, as previously shown, the chemical composition of VIP induced an alteration of the spatial conformation of the CP when using genetic fusion (even with a flexible linker between VIP and CP), thus preventing CPs from self-assembling [28]. A recent study about the viability of TuMV VLPs’ functionalization through genetic fusion has demonstrated, thanks to complex computational analyses, that VIP presents several characteristics that make it a bad candidate to be fused to the CP. Given its α-helix structure and its high proportion of positively charged residues, VIP causes the N-terminus domain of the CP to stretch away from the structural core, giving a non-viable spatial arrangement as a result [30]. Consequently, an alternative technology where VIP and CP were physically more separated and/or the binding of VIP took place after the assembly of the capsid was necessary in order to obtain viable eVLPs. In this context, we considered the SpyTag/SpyCatcher approach for VLPs’ conjugation to peptides as a better option.

The SpyTag/SpyCatcher technology has drawn attention in the last years as a robust functionalizing tool for proteins thanks to its irreversibility and high efficiency. Since its creation, the SpyTag/SpyCatcher approach has been used in several nanotechnological applications based on protein covalent coupling [18,31,32]. One of these applications is VNP functionalization, a field where this technology has been recently implemented with promising results [4]. In this study, we show how the SpyTag/SpyCatcher approach is a successful technique for TuMV VLPs’ functionalization as it allowed us to coat the VLPs with VIP after several unsuccessful attempts via genetic fusion and chemical conjugation [28]. This means that, by using the SpyTag/SpyCacther approach, we circumvented the two main problems encountered with the two aforementioned functionalization methods: the low efficiency of chemical conjugation and the physicochemical constraints of genetic fusion [28]. The successful coating of the VLPs with VIP was only demonstrated when VIP was linked to SpyCatcher, but not in the case with SpyTag. Transmission electron microscopy showed the correct assembly of the VLPs in both co-agroinfiltrations, but Western blot results did not confirm neither the bonding between SpyCatcher-CP and SpyTag-VIP nor the production of SpyTag-VIP alone. The latter may be a consequence of the relatively small size of SpyTag-VIP (6.3 kDa), which may have cause it to run off the SDS-PAGE gel before we stopped the electrophoresis or even to be degraded in the plant prior to protein extraction.

In any case, by functionalizing the eVLPs of TuMV with VIP, we have developed a novel nanotool that could potentially display around 2000 copies of this peptide on its surface. This characteristic make the TuMV-VIP eVLPs attractive nanoparticles with possible future applications in the research of several autoimmune and inflammatory diseases given the role of VIP in this kind of disorders [26,33].

Another interesting result of our study is the demonstration of the isopeptide bond between SpyTag and SpyCatcher occurring *in planta* in both GFP and VIP constructs. In vivo protein conjugation optimizes the functionalization process because in vitro conjugation is more time- and material-consuming (twice as many plants, more crude extracts, and the optimization of the conditions of the in vitro bonding). In our experiment with GFP, we compared *in planta* bonding with a first approach of an in vitro conjugation. The conditions under which the preliminary in vitro reaction was tested were not sufficiently suitable for efficient conjugation, compared with other studies previously published [34,35,36,37]. However, once we checked that the isopeptide bond between SpyTag and SpyCatcher took place *in planta*, the line of in vitro conjugation was not followed. The same outcome regarding *in planta* bonding was previously shown in icosahedral viruses, where VLPs derived from Hepatitis-B virus were functionalized via SpyTag/SpyCatcher [20]. Nevertheless, the occurrence of this conjugation in vivo using flexuous viruses remained unknown. In another study, researchers successfully applied the SpyTag/SpyCatcher technology for the functionalization of the flexuous potato virus X with a *Trichoderma reesei* endoglucanase but, in this case, the binding between SpyTag and SpyCatcher took place in vitro [38]. With our results, we showed how this spontaneous binding of SpyTag and SpyCatcher in vivo is also feasible with flexuous eVLPs of TuMV.

In conclusion, the SpyTag/SpyCatcher technology represents an efficient and cost-saving alternative for the functionalization of TuMV VLPs with proteins and peptides. It allows a straightforward covalent conjugation with the CP *in planta* even in constructs that turned out to be non-viable with previous methods such as chemical conjugation or genetic fusion. The functionalization of TuMV VNPs via this protein coupling technique looks promising for the development of numerous novel nanotools using different proteins and peptides of interest.

## Figures and Tables

**Figure 1 viruses-15-00375-f001:**
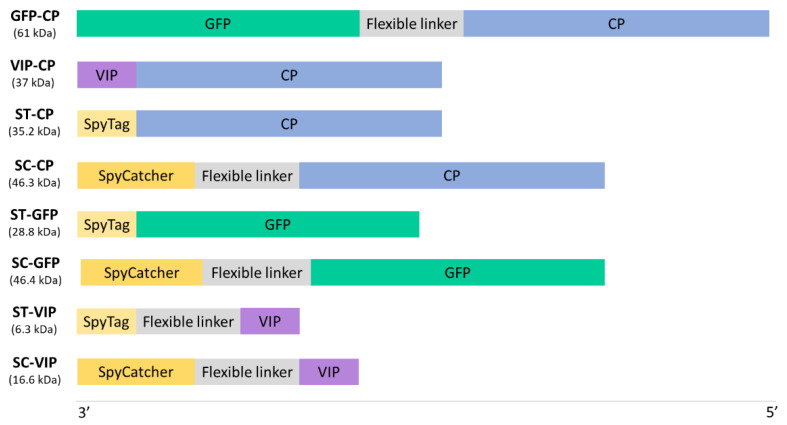
Schematic representation of the eight gene constructs created in this study. A flexible linker (GGGGSGGGGSGGGGS) was added to physically separate the two parts of the fusion protein in those constructs where steric hinderance may happen. ST: SpyTag; SC: SpyCatcher; CP: TuMV capsid protein; GFP: green fluorescent protein; VIP: vasoactive intestinal peptide.

**Figure 2 viruses-15-00375-f002:**
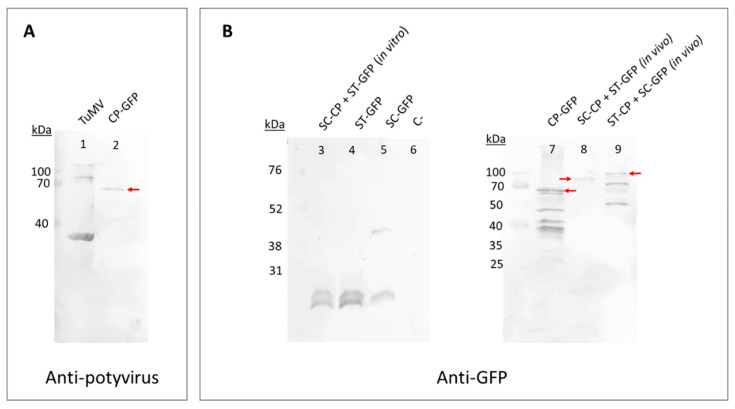
Western blot analyses carried out with the crude extracts of plants infiltrated with the different constructs of VLPs-GFP using the SpyTag/SpyCatcher conjugation technology and genetic fusion. Samples analyzed using anti-potyvirus as the primary antibody are in panel (**A**) and samples analyzed using anti-GFP as the primary antibody are in panel (**B**). For the co-agroinfiltrated constructs, we specify in parentheses if the reaction took place in the plant (in vivo) or in an Eppendorf tube (in vitro). The red arrows highlight the expected band if SpyTag and SpyCatcher are bound (81.6 kDa for SpyTag-CP + SpyCatcher-GFP, and 75.2 kDa for SpyCatcher-CP + SpyTag-GFP) or if the recombinant protein CP-GFP is produced (61 kDa). ST: SpyTag; SC: SpyCatcher; CP: TuMV capsid protein; GFP: green fluorescent protein; C-: Crude extract from a non-agroinfiltrated *Nicotiana benthamiana* leaf.

**Figure 3 viruses-15-00375-f003:**
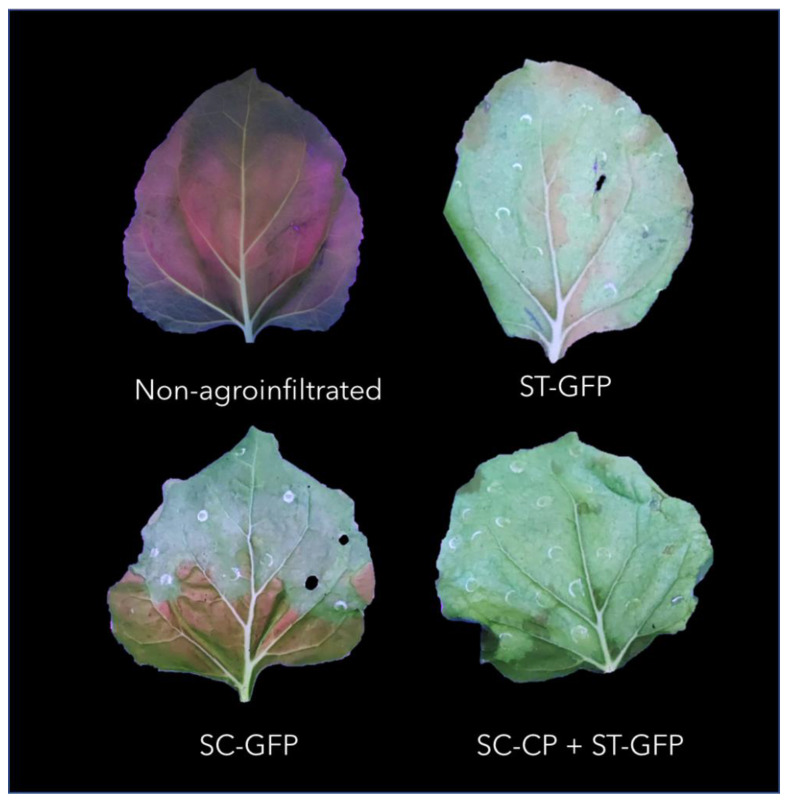
Agroinfiltrated leaves with different VLPs constructs under UV light compared with a non-agroinfiltrated leaf. ST: SpyTag; SC: SpyCatcher; CP: TuMV capsid protein; GFP: green fluorescent protein.

**Figure 4 viruses-15-00375-f004:**
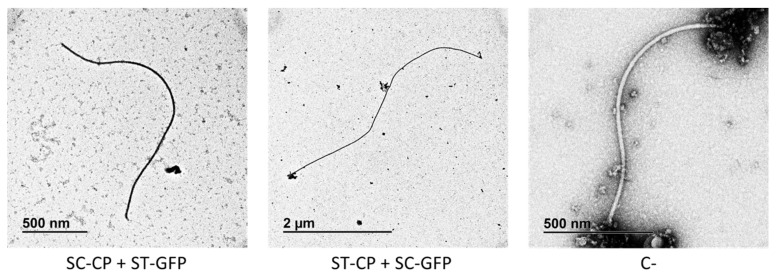
Representative TEM micrographs of the two co-agroinfiltrated constructs of TuMV VLPs-GFP using the SpyTag/SpyCatcher conjugation technology (SC-CP + ST-GFP and ST-CP + SC-GFP). VLPs treated with primary anti-GFP antibody are on the left and central panels. ST: SpyTag; SC: SpyCatcher; CP: TuMV capsid protein; GFP: green fluorescent protein; C-: negative control of an eVLP (ST-CP + SC-GFP) without treatment with the primary anti-GFP antibody.

**Figure 5 viruses-15-00375-f005:**
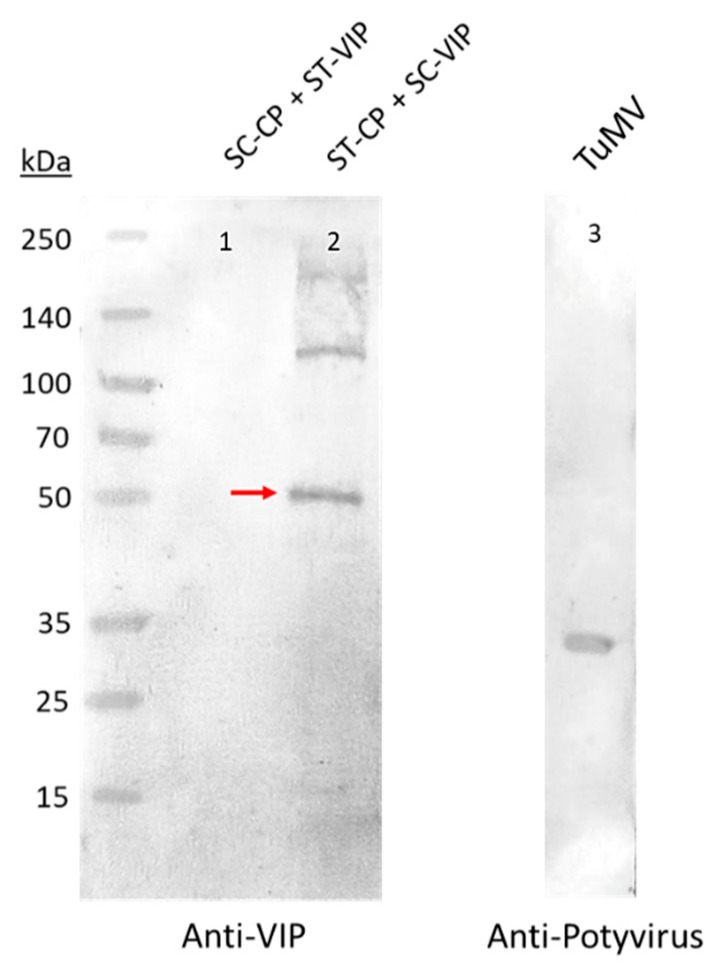
Western blot analyses carried out with crude extracts of plants infiltrated with the different constructs of TuMV VLPs-VIP using the SpyTag/SpyCatcher conjugation technology. The red arrow points to the expected band if SpyTag and SpyCatcher are bound (51.8 kDa). ST: SpyTag; SC: SpyCatcher; CP: TuMV capsid protein; VIP: vasoactive intestinal peptide.

**Figure 6 viruses-15-00375-f006:**
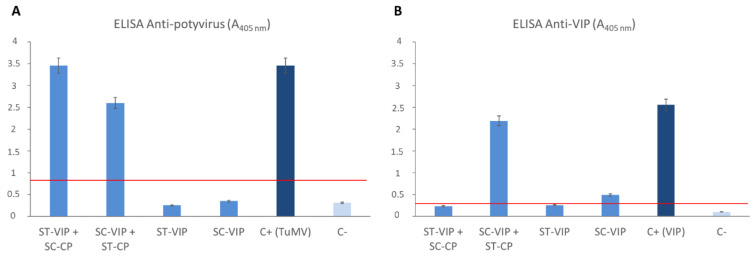
Observed absorbance values at 40 min from ELISAs carried out using anti-potyvirus or anti-VIP as primary antibody with the crude extracts of the different constructs of TuMV VLPs-VIP. The red bar indicates the threshold to consider a positive result (three times the value of the negative control). The error bars show the 95% confidence interval for the absorbance values. ST: SpyTag; SC: SpyCatcher; CP: TuMV capsid protein; VIP: vasoactive intestinal peptide; C+: positive control; C-: negative control.

**Figure 7 viruses-15-00375-f007:**
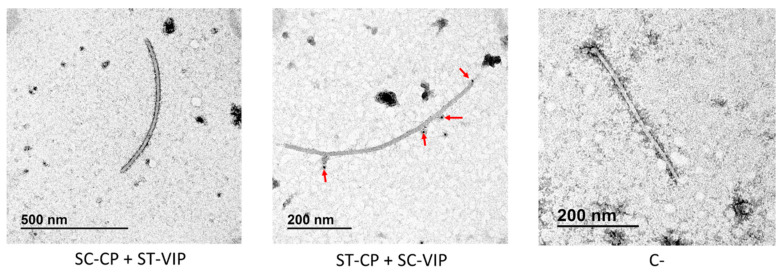
Representative immunogold electron microscopy micrographs of the two co-agroinfiltrated constructs of TuMV VLPs-VIP using SpyTag/SpyCatcher technology. Both constructs were decorated with anti-VIP as a primary antibody and gold-labelled anti-rabbit for decoration. The red arrows highlight the gold nanoparticles attached to the VLPs. Interaction between eVLPs with anti-VIP antibody could only be demonstrated in the ST-CP + SC-VIP construct (central panel). ST: SpyTag; SC: SpyCatcher; CP: TuMV capsid protein; VIP: vasoactive intestinal peptide; C-: negative control of an eVLP without immunogold decoration.

## Data Availability

Data sharing is not applicable to this article.

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
