# Peer review of "Isopeptide Bonding In Planta Allows Functionalization of Elongated Flexuous Proteinaceous Viral Nanoparticles, including Non-Viable Constructs by Other Means"

_viruses, 2023, doi:10.3390/v15020375_

Round 1

Reviewer 1 Report

This paper describes displaying GFP and VIP (vasoactive intestinal peptide) on the surface of Turnip mosaic virus using SpyTag/SpyCatcher. Overall this is a very sound paper in which the conclusions are clearly well supported by the data. A few points indicated below could improve the manuscript, however:

-Line 118: I think the Agrobacterium strain that the authors are referring to is LBA4404 – please double-check.

- Line 183: the authors indicate Figure 3B lane 7 where I think they mean 2B lane 7.

-Figure 6: it is a bit strange to see a bar graph with no error bars. I would like to see error bars, even if they just represent standard error for technical reps of a single experiment (or better yet, 95% confidence intervals from multiple biological replicates of the experiment!)

- Figure 7 needs a bit more explaining, I think. There is much higher contrast in the image on the left (SC-CP + ST-VIP) compared to the one on the right (ST-CP +SC-VIP), which makes it look like the former is covered in gold while the second one is not. This is inconsistent with the data shown in Figure 6. But I think the dark lining of the eVLP shown on the left isn’t gold but simply better UA staining than the image on the right. This confusion isn’t helped by the slightly different magnifications of each image. I recommend that the authors clear up this confusion by stating which of the two images shows more anti-VIP antibody binding and whether this is consistent with Figure 6, and possibly also pointing out each gold nanoparticle with an arrow.

- It would be nice to have some idea of yield of conjugated protein that can be obtained by this method. For example, with in vivo – conjugated ST-CP + SC-VIP, do the authors have any idea what the purified yield might be in terms of mg of purified conjugated eVLP per gram of infiltrated leaf tissue? And how does this compare to the yield of TuMV eVLP alone (i.e. without modifications of conjugations)?

Reviewer 2 Report

In this study, the authors report for the first time the in vivo functionalization of TuMV VLPs surface with GFP or vasoactive intestinal peptide by SpyTag/SpyCatcher technology. SpyTag/SpyCatcher technology can be a good alternative to functionalize viral particles and VLPs with protein molecules in particular in cases when genetic fusion or chemical conjugation is not appropriate. The weakness of this work is the lack of quantitative data on the functionalization of the TuMV VLPs surface by target molecules. Although the results of the study are interesting and may be useful in the development of numerous novel nanotools, the manuscript needs to be revised substantially, including the description of the data.

Major points:

1) If possible, add several sentences in the Introduction or Discussion section reviewing the advantages of TuMV over widely used TMV and PVX.

2) Materials and methods, lines 170-174. The protocol of immunogold electron microscopy should be provided in detail (the presence or absence of the blocking stage, time of incubations, number of washes from antibodies, the size of colloidal gold, etc.).

3) It would have been good to see in the Results section the information about the yield of VLPs functionalized with GFP or VIP.

4) Figure 4, Figure 7. If these are representative images of VLPs, the words “representative micrographs” should be added to the corresponding figure captions.

5) Figure 5. Why are there no lanes for samples SC-CP + ST VIP and ST-CP + SC-VIP analyzed with anti-potyvirus antibodies?

6) The immunogold electron microscopy experiments lack proper controls.

7) In the caption for Figure 6, the number of repetitions of the experiment should be indicated, and error bars should be represented on this figure.

8) Results, section lines 271-272. The authors should clarify the phrase “Immunogold electron microscopy showed that both co-agroinfiltrations were marked with colloidal gold but with different intensity and extension (Figure 7)”. It seems that there is no any interaction of antibodies with VLPs represented on the left part of the figure (SC-CP + ST-VIP).

9) The authors several times mention in the text that SpyTag/SpyCatcher technology offers the high-yield functionalization of TuMV (line 285, lines 318-319, line 348-349). However, without any quantitative data or references, such sentences are unsupported. The authors should provide quantitative data or rephrase these sentences and add appropriate references where necessary.

10) It would be interesting to see in the discussion section the
authors' assumptions on, why the attempt to mix SC-CP+ST GFP in vitro was unsuccessful. It would also be interesting to see some discussion of advantages of using this SpyTag/SpyCatcher technology in vivo over in vitro.

Minor points:

1) If possible, provide the sequence of the flexible linker.

2) Materials and methods, line 110: “form” should be “from”.

3) Materials and methods, section 2.3, line 160. Please, add the binding capacity of plates (high binding or medium).

4) Results, section 3.1, line 183: “and 3B lane 7” should be “and 2B lane 7”.

5) References. The authors should format the reference 30 according to MDPI reference list and citations style guide: Author 1, A.B.; Author 2, C. Title of Unpublished Work. Abbreviated Journal Name year, phrase indicating stage of publication (submitted; accepted; in press).

Round 2

Reviewer 2 Report

The authors have addressed almost all points, and I recommend publication of the revised manuscript after minor changes.

The authors should clarify what does mean "...without decoration" in the caption for Figure 4 (C-).

The authors should specify what is shown in Figure 4. Is it just transmission electron microscopy or TEM of VLPs treated with primary anti-GFP antibodies (SC-CP+ST-GFP; ST-CP+SC-GFP)? If primary antibodies were used in the experiments shown on the left and centre part of the figure (SC-CP+ST-GFP; ST-CP+SC-GFP), the authors should indicate this in the caption.
